# The pornography use and its addiction among emerging adults in Malaysia: Perceived realism as a mediator

**Soon Aun Tan**[1,2]*, **Samantha Hui Li Ng**[1], **Juliana Ju Yun Hoo**[1], **Su Wan Gan**[1], **Sarvarubini Nainee**[1], **Chin Choo Yap**[3], **Ling Khai Lee**[4], **Norzarina Mohd Zaharim**[2], **Yee Shan Goh**[1]

1 Department of Psychology and Counselling, Faculty of Arts and Social Science, Universiti Tunku Abdul Rahman, Kampar, Perak, Malaysia, 2 School of Social Sciences, Universiti Sains Malaysia, Penang, Malaysia, 3 Department of Psychology, School of Medical and Life Sciences, Sunway University, Selangor Darul Ehsan, Malaysia, 4 Department of Languages and Linguistics, Faculty of Arts and Social Science, Universiti Tunku Abdul Rahman, Kampar, Perak, Malaysia

* tansa@utar.edu.my

**Data Availability Statement:** All relevant data are within the paper and its Supporting Information files.

## Abstract

Past studies have demonstrated a link between pornography use and addiction to pornography, the underlying mechanism of the association is still unclear. This study intended to examine the mediating role of perceived realism of pornography in the association between pornography use and addiction among emerging adults in Malaysia. This study recruited 222 self-identified pornography users (M age = 21.05; SD Age = 1.68; 75.1% male respondents) via the purposive sampling method. The correlation results indicated positive associations among pornography use, addiction to pornography, and perceived realism of pornography. Further, the perceived realism of pornography significantly mediated the association between pornography use and addiction to pornography after controlling for gender. Thus, this study has provided a fundamental understanding on the perceived realism role of pornography in explaining the association between its use and addiction. Although it is unlikely to stop illegal pornography use, the results pointed out a need to guide emerging adults in pornography use via media literacy programmes.

## Introduction

Given the rise in usage of electronic devices and the Internet, pornographic materials have been made easily accessible online [1, 2]. A popular pornography website, PornHub, reported an estimated 39 billion searches and 42 billion visits to the site in the year 2019. These numbers are estimated to be an equivalent of 115 million visits per day in the same year [2]. This data shows the extent of the demand and consumption of pornography nowadays.

Past studies in the Australian [3] and a global report [2] on the gender profile of pornography users in developed nations reported that around 46–76% of males are pornography users while 16–41% are females. Rissel et al. [3] found that 4% out of the 76% of active male pornography users were addicted to pornography, while 1% out of the 41% active female pornography

**Funding:** The research has been carried out under the Fundamental Research Grant Scheme project FRGS/1/2019/SS05/UTAR/03/1 provided by the Ministry of Higher Education of Malaysia.

**Competing interests:** The authors declare that they have no conflict of interest.

users were addicted to pornography. Moreover, a total of 20% of emerging adults are regular monthly users of pornography, while 15% view pornographic materials daily [4]. Although pornography use can stimulate the sexual desire of couples as well as strengthen intimate relationships and bonding between them [5, 6], its excessive use could also bring adverse effects to emerging adults' physical health and well-being. During emerging adulthood, pornography addiction may affect individuals' cognition and drive for sexual stimulation. Individuals who are addicted to pornography display obsessive thoughts about sex and have a compulsion to view sexually explicit materials repeatedly thus leading to sleep disorders [7], depression [7, 8] and difficulties in developing interpersonal relationships with others [7, 9].

## Pornography use and pornography addiction

Pornography refers to two components which are sexually explicit materials and sexual intention. Sexually explicit materials include nudity and sexual behaviour [10, 11], while sexual intention refers to the intention to increase sexual arousal among users [10]. Thus, pornography use can be defined as the frequency of media consumption depicting nudity and sexual acts for sexual arousal and excitement. An addiction has been defined as "the continued use of mood-altering, addictive substances or behaviours (e.g., gambling, compulsive sexual behaviours) despite its adverse consequences" ([12] p. 696). Therefore, pornography addiction can be defined as the "excessive and uncontrollable consumption of pornography despite its negative consequences".

Grubbs et al. [13] found that daily pornography use is a predictor of self-reported pornography addiction, especially among male users. Pornography use has been shown to cause pleasure and a sense of reward as feel-good chemicals such as dopamine are released when viewing pornography [14]. Pornography use can reinforce the individual's habit of watching more pornography to attain a feeling of excitement from pornographic materials. Individuals who constantly watch pornography are predisposed to build tolerance and may seek out more intense pornography to maintain or achieve higher levels of sexual excitement, thus leading to compulsive pornography consumption or addiction [14].

## Pornography use and perceived realism

Perceived realism refers to the extent to which its viewers perceive material as real [1, 15]. The content of a material may be viewed as realistic fiction on two criteria which are external realism and narrative realism. External realism denotes the fit between the story and real-life, while narrative realism refers to the consistency and intelligibility of the story [16, 17]. In pornography use, perceived realism refers to how real an individual perceives the pornographic materials to be. Individuals with a higher level of perceived realism of pornography had a higher tendency to imitate sexual behaviours seen in pornographic materials as they were motivated to achieve the same pleasure responses they felt when watching pornography [1, 15, 18, 19]. Past studies have found that higher pornography use led to a higher perceived realism of pornography. Therefore, the individual with high perceived realism of pornography reported having a greater level of unrealistic expectations on sex because the individual had mistakenly believed that the real-life relationships and sexual activity would be similar as had been portrayed in pornographic materials [1, 20].

The existing literature primarily focused on adolescents' perceived realism and pornography use [19–22]. Nevertheless, the needs to seek knowledge regarding the perceived realism and pornography use among emerging adults, are overlooked. As the relevant literature on the emerging adult population is limited, this study reviewed only past studies on adolescents. A longitudinal study by Wright and Štulhofer [19] had shown the changes in pornography use

and the perceived realism of adolescents over time. During the transition period from middle to late adolescence, there was a significant increase in pornography use and a decrease in perceived pornography realism during late adolescence which stabilised after age 17. However, the perceived realism and experiences of pornography viewing could be different between adolescents and emerging adults. Valkenburg and Peter [21] suggested that individuals' beliefs vary depending on their exposure experiences. The distinctive cognitive, emotional and excitement responses demonstrated by adolescents and emerging adults explain how and why media influences these individuals differently. According to the differential susceptibility to media effects model, individuals selectively pay attention to media content and interpret it differently. Also, the emotional and excitement response states suggested that individuals possessed different affective reactions and experienced different degrees of psychological arousal towards the media content respectively [21]. Furthermore, past studies on perceived realism were conducted in Western countries where pornography and sexual activity are normative among adolescents [22–24]. Therefore, it is important to examine the linkage between pornography use and the perceived realism of pornography among emerging adults, particularly in a conservative society such as Malaysia.

## Perceived realism of pornography and pornography addiction

According to Busselle and Bilandzic [16], if an individual views the content of materials as realistic, the individual tends to engage more in that particular material. The increased engagement leads the individual to experience a flow-like state, causing them to be unaware of their surroundings and lose self-awareness [16, 25]. Hence, individuals who deem pornography realistic are likely to be more immersed or engaged in the pornographic material [26]. The high perceived realism of pornography may result in one being engrossed with pornography to the point of losing self-awareness and self-control; this, in turn leads to a higher risk of developing pornography addiction as the individual is compelled to keep watching pornography to maintain the same level of pleasure and sexual excitement [27].

Previous studies had found that pornographic fantasies and interests were positively associated with problematic use of online sexual activities, including viewing pornography [28, 29]. Due to social taboo, sexual fantasies and interests are considered as socially inappropriate thoughts which are unacceptable for most people. Thus, to satisfy the sexual fantasies which cannot be executed in the real world, some individuals may seek out unrealistic pornographic content in the virtual world [25, 30]. As sexual fantasies can be fulfilled through pornography use, the individual tends to constantly crave and use pornography more in order to achieve sexual excitement unattainable in real life [14, 27, 30]. Over time, perceived realism that can fulfil the individual's unrealistic sexual interest may develop into excessive pornography use or addiction.

On the other hand, Ali et al. [1] conducted a quantitative study on Malaysian college students and found that they were less likely to be involved in problematic use of pornography if they considered sexually explicit materials as real. This is because viewing pornography and premarital sex activities are prohibited in the Malaysian society, especially among the vast religious communities, which regards the non-marital sex as a sin. Therefore, those who perceive pornography as real will avoid the behaviour, and thus decrease pornography addiction.

## Perceived realism as a mediator

The studies on the mediating role of perceived realism in the literature pool is mostly based on adolescents and the studies among emerging adults are limited [20, 22]. The increased in pornography usage will increase the extent to which adolescents perceive the sexual scripts in

pornography as similar to the real-world sex [20]. The teenagers evaluate the reality of pornography by comparing the sexual scripts shown in pornography and their own sexual scripts. Adolescents who use pornography more frequently tended to synchronise their own sexual scripts with sex depicted in pornography as a result of their perceived realism of pornography use [20]. The findings from the study of Vandenbosch et al. [22] revealed that pornography use contributed to pornography addiction by increasing the perceived realism of individuals. Individuals who view sexually explicit material as real tended to adopt the sexual scripts from pornography into their real life. A study by Chock [31] also found that individuals who viewed pornography as real and perceived its usefulness were more likely to adopt the sexual scripts demonstrated in pornography into their real life.

In addition, positive feedback from the partners on the individuals' sexual performance will cause these individuals to engage in an even more extensive use of pornography. This is expected and reasonable especially if they found that the behaviours exhibited in pornography are rewarding and applicable in their real-life setting; thus, they are more likely to perceive pornography to be real and practical [32]. In this case, individuals who view pornography scripts as real will be more inclined to value the positive effects (i.e., the pleasure aroused from watching pornography) of viewing pornography over the negative effects [33]. Therefore, pornography use of individuals may lead to a higher level of pornography addiction via the increasing level of perceived realism on the sexual scripts in pornography. Therefore, emerging adults who use pornography are more likely to perceive pornography materials as real which will lead them to continue seeking sexual sensations from pornography and risking of becoming addicted to pornography.

## The present study

This study was conducted to enrich the literature regarding the association of perceived realism, pornography use and pornography addiction within the Malaysian context. There are limited local studies that are focused on emerging adults [1]. Hence, the present study focuses on emerging adults. The present study delves further by examining the associations of pornography use, perceived realism and pornography addiction among emerging adults in Malaysia. The exiting literature has mainly focused on pornography use and perceived realism, with very few studies focusing precisely on the associations between perceived realism and pornography addiction [15].

Furthermore, the present study looks into the mediating role of perceived realism of pornography use and its association between pornography use and addiction. As most previous research on this linkage has been conducted in western societies, the present study could enrich the literature on sexuality, particularly in pornography use in the Malaysian context. Moreover, despite numerous past studies, the mediating role of perceived realism on pornography use and pornography addiction is not sufficiently researched. Therefore, the present study aimed to fill in the research gaps by answering the two research questions below:

1. Does pornography use and perceived realism each link with pornography addiction among emerging adults in Malaysia?

2. Does perceived realism of pornography use mediate the association between pornography use and pornography addiction among emerging adults in Malaysia?

## Method

### Participants

This study involved 222 emerging adults aged between 18 and 27 years old with a mean age of 21.05 and a standard deviation of 1.68. More than three-quarters of the respondents were male

(75.1%). In addition, 82% of the respondents were Chinese, followed by 8.6% Malays, 6.3% Indians, and 3.2% of them are from minority groups. A total of 67.1% of the participants were single, 29.7% were currently in a relationship with the opposite sex, 2.7% were currently in a relationship with the same sex, and 0.5% were currently married.

## Procedure

An online survey method was used to collect the data of this study. The respondents of this study were recruited using the purposive and snowballing sampling methods. The respondents must be Malaysians emerging adults aged 18 to 29 and self-identified as pornography users. The online survey link was posted on various social media platforms (i.e., Facebook, Twitter, and Telegram). Firstly, the respondents were briefed on the research objectives, participants' rights, potential risks and benefits of joining this study, and the protection of their privacy and confidentiality. They were required to indicate their written consent before answering the questions. The UTAR Scientific and Ethical Review Committee has approved the research procedure and data collection (Ref: U/SERC/08/2020).

## Measures

**Cyber Pornography Use Inventory (CPUI) [34].**   CPUI was used to measure pornography addiction. This scale contains nine items to measure compulsivity, effort, and distress with three items each. Respondents were asked to rate on the rating scale from 1 (*Not at all*) to 7 (*Extremely*). The present study utilised the unidimensional score to assess pornography addiction. A mean score was computed with a higher score corresponding to a higher level of pornography addiction. The Cronbach alpha reliability was .86 which indicates excellent internal reliability.

**Perceived realism of sexually explicit internet materials scale [20].**   Perceived realism was measured using the scale developed by Peter and Valkenburg [20]. This scale consists of seven items that the respondents have to rate on a 5-point Likert scale ranging from 1 (*strongly disagree*) to 5 (*strongly agree*). A mean score was computed with a higher score corresponding to a higher perception of the realism of pornography use. The Cronbach alpha reliability was .81 which indicates excellent internal reliability.

**Pornography use.**   A single item was developed to ask about respondents' pornography use in the past month. Respondents were required to indicate their engagement based on a 5-point rating scale from 1 (*Not at all*), 2 (*once or a few times*), 3 (*once a week*), 4 (*a few times a week*) and 5 (*every day or almost every day*).

## Results

Table 1 describes the descriptive statistics and correlation results between pornography use, perceived realism and pornography addiction. Results revealed a positive association between pornography use and perceived realism. Likewise, perceived realism positively correlated with pornography addiction. The individual with a higher pornography use also reported a higher level of pornography addiction.

Hayes' PROCESS Macro [35] for SPSS version 4.0 was used to perform mediation analysis (Model 4) to examine the indirect effect between pornography use and pornography addiction via perceived realism of pornography use. Gender was preserved as covariance in the present study. The 95% bias-corrected confidence interval (CI) was generated using 10,000 bootstrap samples. A significant indirect effect is detected if both CIs are in the same direction.

The result of the study found that pornography use was positively correlated with pornography addiction (total effect model), $B = .25$, $SE = .07$, $t (218) = 3.62$, $p < .001$, 95% $CI$ [.11, .38].

**Table 1. Cronbach alpha reliability, descriptive and matric correlation between variables ($n$ = 222).**

|  | Cronbach Alpha | 1 | 2 | 3 |
|---|---|---|---|---|
| 1. Pornography Use | - | 1 |  |  |
| 2. Perceived Realism | .81 | .28*** | 1 |  |
| 3. Pornography addiction | .86 | .24*** | .44*** | 1 |
| Mean |  | 3.28 | 2.62 | 3.37 |
| SD |  | 1.17 | 1.18 | .91 |
| Skewness |  | .34 | 1.18 | .04 |
| Kurtosis |  | -1.37 | 1.75 | .30 |

*Note.*

*** $p$ < .001.

Pornography use remained significantly associated with pornography addiction after controlling the effects of perceived realism and gender (covariance), $B$ = .14, $SE$ = .06, $t$ (217) = 2.18, $p$ = .03, 95% $CI$ [.01, .27]. Meanwhile, pornography use was positively linked to perceived realism, $B$ = .20, $SE$ = .05, $t$ (218) = 3.80, $p$ < .001, 95% $CI$ [.10, .31]. Similarly, perceived realism was also positively linked to pornography addiction, $B$ = .52, $SE$ = .08, $t$ (217) = 6.65, $p$ < .001, 95% $CI$ [.37, .68]. Gender was found to be significantly associated with pornography addiction, $B$ = .42, $SE$ = .17, $t$ (217) = 2.50, $p$ = .01. The mediating effect of perceived realism was significant in its association between pornography use and pornography addiction, $B$ = .11, $SE$ = .04, 95% $CI$ [.04, .19]. Refer to Fig 1 for the associations between the variables.

## Discussion

The results revealed a positive association between pornography use and pornography addiction. Likewise, perceived realism positively correlated with pornography addiction. Also, the present study revealed that perceived realism mediates the association between pornography use and pornography addiction.

The results suggested that pornography use is positively correlated with higher perceived realism among Malaysian emerging adults. This means that participants who claimed to use pornography more often were more likely to treat the sexual scripts shown in pornography as real. Baams et al. [15] using the social cognitive theory posited that adolescents are more likely

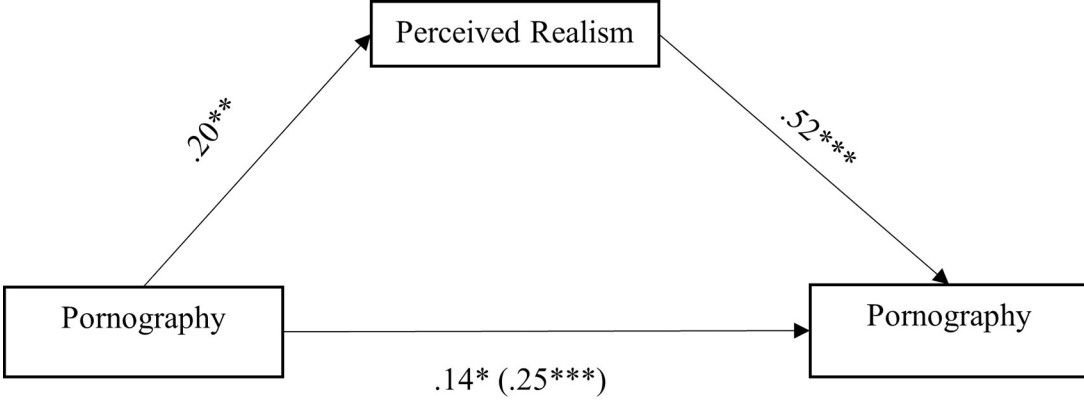

**Fig 1. A mediation model showing the effect of pornography use and perceived realism on pornography addiction.** The Values Shown are Unstandardised Coefficients. The Total Effect was Shown in Parenthesis. *$p$ < .05; ***$p$ < .001.

to watch sexually explicit materials that match their level of acceptance as well as seeing the pornography materials as a social model. Although adolescents and emerging adults could have different experiences in pornography use and perceived realism, the current study also found a significant association between pornography use and perceived realism among emerging adults, consistent with the study by Baams et al. [15] on adolescents. In addition, our findings were also supported the study by Vogels [18] which suggested that greater use of pornography was associated with greater perceived realism among adults aged between 19 and 30.

Furthermore, the results posited a positive correlation between pornography use and pornography addiction. When individuals experience excitement from sexually explicit materials for the first time, they tend to develop pornography addiction to seek higher levels of sexual excitement [14]. A biological perspective also supported this correlation. The findings by Kühn and Gallinat [36] on 64 male participants revealed that a longer duration of pornography consumption was associated with reduced grey matter volume in the right caudate. The decrease in grey matter volume disrupts the rewarding process in the brain, causing the individual to lose control and subsequently become addicted [37]. Also, the MRI study done by Kühn and Gallinat [36] showed that higher pornography use was correlated with lower left putaminal activity in response to static sexual images. In other words, the lowered putamen activity caused desensitisation, whereby individuals will develop tolerance towards sexual stimuli [36]. Thus, the individuals tend to spend more time on pornographic materials to gain the same sexual stimuli after their first exposure thus leading to pornography addiction in the future.

Perceived realism is also related to pornography addiction. Perceiving sexual material as real evokes individuals' engagement in further pornography consumption [19]. The metacognitive model suggested that positive metacognitions about desired thinking directly impacted individuals' imaginal prefiguration (i.e., mental imagery of desired behaviour), causing them to recall or visualise their experiences watching pornography [30]. Thus, individuals will continue to plan and engage in the desired behaviour. Taking it altogether, the experience of pornography use can make emerging adults perceive pornographic content as a more realistic event and visualise the close-to-life sexual scripts from pornography that can arouse them to engage in excessive use of pornography. Previous studies also found that pornography use may cause the individual to lose control of their behaviour which leads to pornography addiction [30, 38].

Nevertheless, the results are inconsistent with the findings from the study by Ali et al. [1] which indicated that Malaysian college students who were exposed to pornography use and perceived it as real did not exhibit pornography addiction. In other words, Ali et al. [1] posited that perceived realism did not mediate the association between pornography use and pornography addiction. Viewing pornography is considered sinful among the main religions in Malaysia especially the Muslim community and Islam being the most practised faith. Thus, they will avoid being exposed to sexually explicit materials when they perceive pornography to be real. The potential discrepancy between the study by Ali et al. [1] and the present study might be due to participants' demographic backgrounds. The participants in the study of Ali et al. [1] were mainly Malays, with up to 90.8%, whereas this study recruited 82% Chinese and only 8.6% Malay participants. Moreover, Jafarkarimi et al. [39] affirmed that Malaysian college students confirmed that pornography use is affected by religion where Muslims are less likely to view pornography than non-Muslims. Therefore, cultural factors across ethnic and religious backgrounds could explain the differences between perceived realism and pornography addiction. Future studies may address this potential gap by considering ethnic and religious factors in understanding the link between perceived realism and pornography addiction.

The findings also revealed a significant mediation model in the indirect relationship between pornography use and pornography addiction via perceived realism. Malaysia's emerging adults who were exposed to pornography more frequently developed greater perceived realism of the sexual scripts and were subsequently involved in problematic use of pornography. Consistent with the present study, Wright and Štulhofer [19] posited that individuals with more frequent pornography use tended to perceive pornography materials as more real. Therefore, the ingrained perceived realism caused them to continuously seek sexual pleasure from pornographic materials [33]. The aforementioned positive metacognitions of desired thinking about pornographic materials caused the individuals to develop pornography addiction when they perceive the sexually explicit materials as more real than reality.

## Limitation and recommendations

Nevertheless, we acknowledge that this study has several limitations. Firstly, the generalizability of this study is a constraint as the researchers adopted the non-probability sampling method due to the sensitiveness of the topic studied. In addition, this study may have recruited a biased sample as the sample was self-selected. The questionnaire was advertised with the topic of pornography. As a result, the study may potentially attract those with sexual problems, leading to an overestimation of pornography consumption.

Secondly, this study does not fairly reflect Malaysian public as the ethnic composition of the participants did not correspond to the ethnic ratio in Malaysia, with the Chinese participants in this study being approximately ten times more than the Malay participants. Hence, future studies should consider using probability sampling such as strata sampling to recruit participants to balance the composition of ethnicities. Also, a randomised sampling method should be adopted to select the participants rather than attract potential active pornography users.

Thirdly, the researcher adopted a cross-sectional study that limits the identification of the causal direction of the mediating patterns found between pornography use, perceived realism, and pornography addiction. Although the causal link between the variables cannot be proven in this study, the findings can provide preliminary support to the hypothetical mediation model. Future research can replicate the current research using longitudinal research design to confirm the cause-and-effect between pornography use, perceived realism and pornography addiction.

Lastly, each dimension of perceived realism was not studied separately. Instead, perceived realism was considered as a whole to provide a general perceived realism component. This is because not many studies had been conducted in this context. There are different forms of perceived realism as studied in previous research, such as social realism, utility [20], factuality, consistency, narrative consistency, and perceptual quality, plausibility and typicality [40]. Future research should focus on evaluating different dimensions of perceived realism to examine their potentially different effect on pornography viewing.

## Conclusion

The findings of this study have provided a clear picture of the correlation between pornography use and pornography addiction among emerging adults in Malaysia. Moreover, this study found that the association between pornography use and its addiction was mediated by perceived realism. However, the results stress the importance of acknowledging that the sexually explicit materials do not affect all emerging adults in the same way, and how the individuals perceive the materials plays a role as well. Therefore, this study highlights the need for future research to specifically examine the consequences of perceived realism in pornography that

depicts sexual acts deemed immoral or illegal in real life. This study supports the ongoing effort for sex education providers to promote media literacy on unrealistic pornography production to minimise and to reduce perceived realism, thereby minimising the dependency on pornography use. In addition, this study allows future researchers or governmental bodies to better understand pornography use (generally) and the mechanisms leading to pornography addiction among emerging adults (specifically) as most studies involving pornography use and addiction were extensively based on adolescents [41]. Therefore, relevant authorities can take measures best suited to the local needs to promote better porn literacy as it is vital for developing a healthy Malaysian society.

## Supporting information

**S1 Data. 222 DATA pornography study.**
(SAV)

## Author Contributions

**Conceptualization:** Soon Aun Tan, Su Wan Gan, Sarvarubini Nainee, Chin Choo Yap, Ling Khai Lee, Norzarina Mohd Zaharim, Yee Shan Goh.

**Formal analysis:** Soon Aun Tan.

**Funding acquisition:** Soon Aun Tan.

**Investigation:** Soon Aun Tan, Su Wan Gan, Sarvarubini Nainee, Chin Choo Yap, Ling Khai Lee, Norzarina Mohd Zaharim.

**Methodology:** Soon Aun Tan.

**Project administration:** Soon Aun Tan.

**Supervision:** Su Wan Gan.

**Writing – original draft:** Soon Aun Tan, Samantha Hui Li Ng, Juliana Ju Yun Hoo.

**Writing – review & editing:** Soon Aun Tan, Samantha Hui Li Ng, Juliana Ju Yun Hoo, Su Wan Gan, Sarvarubini Nainee, Chin Choo Yap, Ling Khai Lee, Norzarina Mohd Zaharim.

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
