## [Decision Letter · Decision Letter 0]

6 May 2022

THE PORNOGRAPHY USE AND ITS ADDICTION AMONG EMERGING ADULTS IN MALAYSIA: PERCEIVED REALISM AS A MEDIATOR

PONE-D-22-09031

Dear Dr. Tan,

We’re pleased to inform you that your manuscript has been judged scientifically suitable for publication and will be formally accepted for publication once it meets all outstanding technical requirements.

Kind regards,

Halimatus Sakdiah Minhat, DrPH

Academic Editor

PLOS ONE

Additional Editor Comments (optional):

Reviewers' comments:

Reviewer's Responses to Questions

**Comments to the Author**

1. Is the manuscript technically sound, and do the data support the conclusions?

Reviewer #1: Yes

2. Has the statistical analysis been performed appropriately and rigorously? 

Reviewer #1: Yes

3. Have the authors made all data underlying the findings in their manuscript fully available?

Reviewer #1: Yes

4. Is the manuscript presented in an intelligible fashion and written in standard English?

Reviewer #1: Yes

5. Review Comments to the Author

Reviewer #1: Limitation of sample population was addressed as the Malaysia population has a 60% Malay population. This was further suggested to conduct a study that uses probability sampling and randomization. This could be more meaningful as the result can used to generalize the Malaysian emerging adults.

6. PLOS authors have the option to publish the peer review history of their article (what does this mean?). If published, this will include your full peer review and any attached files.

Reviewer #1: No

---

## [Editor Report · Acceptance letter]

12 May 2022

PONE-D-22-09031 

The pornography use and its addiction among emerging adults in Malaysia: Perceived realism as a mediator 

Dear Dr. Tan:

I'm pleased to inform you that your manuscript has been deemed suitable for publication in PLOS ONE. Congratulations! Your manuscript is now with our production department. 

Kind regards, 

on behalf of

Dr. Halimatus Sakdiah Minhat 

Academic Editor

PLOS ONE